# Correlations between Psychiatric Symptoms and Quality of Life in Patients with Psychological Disorders: Hospital-Based Retrospective Study

**DOI:** 10.3390/ijerph19020732

**Published:** 2022-01-10

**Authors:** Pi-Yu Su, Shu-Fen Kuo, Min-Huey Chung

**Affiliations:** 1School of Nursing, College of Nursing, Taipei Medical University, 250 Wuxing Street, Taipei 11031, Taiwan; nursepiyu@gmail.com (P.-Y.S.); sfkuo6@tmu.edu.tw (S.-F.K.); 2Director of Nursing, Penghu Branch, Tri-Service General Hospital, National Defense Medical Center, Penghu 88056, Taiwan; 3Department of Nursing, Shuang Ho Hospital, Taipei Medical University, New Taipei City 23561, Taiwan

**Keywords:** psychiatry, quality of life, mood disorders, adjustment disorders

## Abstract

Little research has been conducted on the relationship between the five-item Brief Symptom Rating Scale (BSRS-5) and quality of life in patients with mood disorders. The purpose of this study was to investigate potential effects of psychological symptoms on quality of life in patients with psychological disorders. We recruited 124 patients with psychological disorders from a psychological teaching hospital in northern Taiwan. Data were obtained from medical records of all patients with a diagnosis of mood or adjustment disorder. We assessed psychological symptoms on the BSRS-5 and examined quality of life by using the Taiwanese version of the abbreviated World Health Organization Quality of Life Questionnaire (WHOQOL-BREF). We performed hierarchical linear regression analysis to explore the relationship between psychological symptoms and quality of life. The analysis revealed a significant correlation between the items on the BSRS-5 and WHOQOL and their correlations with the total scores on these assessments (*p* < 0.01 and *p* < 0.05). Our findings indicated that scores on the BSRS-5 can predict scores on quality of life. This suggests that psychometrically measured psychological symptoms constitute critical determinants of quality of life.

## 1. Introduction

The prevalence of psychological disorders increased by 13% between 2007 and 2017, mainly because of demographic changes. Mood disorders cost the global economy US$1 trillion each year [1]. In 2019, the World Health Organization (WHO) launched the WHO Special Initiative for Psychological Health (2019–2023): Universal Health Coverage for Psychological Health [2]. Mental health conditions can have a significant impact on all aspects of life, including school or work performance, relationships with family and friends, and community participation [1]. Common mood disorders include non-psychotic, depressive, and anxiety disorders. Patients with adjustment disorders may also have emotional symptoms, which are milder than mood disorders. Furthermore, psychological disorders increase the risk of both communicable and noncommunicable diseases, which lead to unintended injury [3]. Psychological disorders rank first in the global burden of disease, equivalent to the combined burden of cardiovascular and circulatory diseases. From a public health perspective, health cannot be achieved without psychological health, as indicated by the fact that poor psychological health increases national unemployment, divorce, and suicide rates [4,5]. Severe psychological diseases can trigger a cascade of socioeconomic and lifestyle changes that, in turn, can lead to adverse physical health effects and increase mortality risk to approximately two to three times that of the average person [6]. Therefore, the examination of both social and psychological measures is essential [4].

Quality of life does not concern only physical and psychological well-being but also social and environmental status [7,8]. In one study, age and employment status were significantly associated with quality of life [9]. In another study, quality of life and desire to die were among the factors used to assess the frequency and characteristics of suicidal ideation in older patients in Taiwan with medical or surgical conditions [10]. Literature shows that both age and education negatively affected quality of life in outpatients with schizophrenia living in France, Germany, and the United Kingdom [11]. Another study has shown that depression and anxiety affect quality of life in the healthy population [7]. Furthermore, a study by Sagayadevan, Lee [9] indicated that mood disorders, such as depression and anxiety, were associated with the lowest quality of life and the highest burden among all psychological disorders examined. Huang, Chen [12] observed that the most critical predictors of quality of life in patients with chronic psychological illness (CMI) were disease factors.

Psychological symptoms, which include insomnia, depression, anxiety, and interpersonal sensitivity, develop when people experience emotional stress or an important life event such as divorce or bereavement [7]. The Five-Item Brief Symptom Rating Scale (BSRS-5), a screening tool used to track psychological disorders in non- psychological health settings [13], can effectively screen patients for mood disorders and general medical conditions. Furthermore, it can screen community residents for suicide vulnerability [13,14]. In general, psychological symptoms can affect clinical signs, physical disease complications, and quality of life [7].

Studies have examined BSRS-5-related factors in elderly in-patients with surgical or medical conditions. However, they did not explore the mental factors related to the suicidal ideation [10]. One study included subjects from community residents with an exclusion of cases of mental illness [14]. Regarding research methods, some studies reported the use of telephone interviews as research methods. In these studies, psychiatric medical staff were not included as researchers to avoid bias [13,14]. In addition, other studies indicated that apart from psychiatric patients, they also examined adolescents, suspected or confirmed coronavirus disease-19 (COVID-19) patients, healthcare workers, and soldiers [13,15,16,17].

Overall, previous studies have rarely explored the correlation between BSRS and quality of life among patients with a mood disorder or adjustment disorder. In this study, the aim was to investigate the potential impact of psychological symptoms on quality of life of patients with a mood or adjustment disorder. Predicting risk factors helps in early detection and in the provision of appropriate intervention programs.

## 2. Methods

### 2.1. Participants and Settings

We used a hospital-based retrospective study design. We included patients older than 20 years with a diagnosis of mood or adjustment disorder using a convenience sampling method. The study was conducted in acute wards at a psychological teaching hospital in northern Taiwan. Data were obtained from the medical records of all patients with a diagnosis of mood or adjustment disorder in accordance with the criteria outlined in the Diagnostic and Statistical Manual of Mental Disorders, Fifth Edition. This study was approved by the Institutional Review Board (approval number N1-107-05-066).

The sample size was calculated using G*Power software (version 3.1.9.4). Power analysis involved a linear multiple regression with a small effect size of 0.2 [18,19], using a power of 80% and a significance level of 5% (two tails). The sample size was at least 42 participants. In this study, the medical records of 124 patients were searched from February 2018 to May 2018.

### 2.2. Information from Medical Records

Data on the patients’ demographic information (sex, body mass index [BMI], age, education, and smoking and drinking status), disease characteristics (age at disease onset and diagnosis), psychological symptoms (as indicated by BSRS scores), and quality of life (as indicated by scores on the World Health Organization Quality of Life (WHOQOL-instrument) were obtained from their medical records.

### 2.3. Five-Item Brief Symptom Rating Scale (BSRS-5)

The BSRS-5, a self-report questionnaire, was modified from the Symptom Checklist-90 Revised at National Taiwan University Hospital by Lee, Liao [20]. It can effectively detect individuals’ psychological care needs through the evaluation of five symptoms, namely; anxiety, depression, hostility, interpersonal sensitivity/inferiority, and insomnia, as experienced over the past week. The BSRS-5 provides a quick understanding of the respondent’s physical and psychological health. Concurrent validity coefficients between total scores on the BSRS-5 and the 50-Item BSRS (BSRS-50) general severity index range from 0.87 to 0.95. From receiver operating characteristic curve analysis, a cut-off score of 6+ points on the BSRS-5 for psychological disorders was calculated. In short, the BSRS-5 is a vital instrument used in the early detection of psychological disorders [13]. Items are scored on a 5-point Likert scale, where 0, 1, 2, 3, and 4 points are given for responses of ‘not at all’, ‘a little bit’, ‘moderately’, ‘quite a bit’, and ‘extremely’, respectively. The internal consistency of the scale, as indicated by Cronbach’s α values ranging from 0.77 to 0.90, is between acceptable and good. The test–retest reliability coefficient is 0.82 [20]. Higher scores on the BSRS-5 indicate poorer psychological health from item 1 to item 5. We assigned the sixth item in BSRS-5 to directly ask about the responder’s current suicidal ideation, ranging from 0 for none to 4 for strong suicidal ideation.

### 2.4. The Abbreviated World Health Organization Quality of Life Questionnaire (WHOQOL-BREF)

In 1991, the WHO launched a project to develop a generic measure of quality of life in 10 countries. This 100-item instrument, called the WHOQOL-100, was finalized in 1995 [21]. In consideration of the fact that a considerable amount of manpower and time is required for the implementation of this instrument in research, the WHO devoted its efforts to developing a concise version. Specifically, research and development in 18 countries has yielded 26 brief versions, encompassing the four domains of physical health, psychological health, social relationships, and environment (scores on which are correlated by ≥0.89). Domain scores on the WHOQOL-BREF, an abbreviated version of the WHOQOL-100, demonstrate favourable discriminant validity, content validity, internal consistency, and test–retest reliability [22].

In 1998, a Taiwanese research group developed the Taiwanese version of the WHOQOL-BREF in Geneva, Switzerland. This 28-item version comprises four domains, namely; physical, psychological, social relationships, environment, and local culture, which are addressed by seen, six, four, nine, and two items, respectively. Responses are scored on a 5-point Likert scale of 0–4 points. Higher total scores indicate better quality of life. Regarding internal consistency, the Cronbach’s α coefficients of the four domains range from 0.70 to 0.77. The coefficients of test–retest reliability at two to four week intervals ranges from 0.41 to 0.79 at the item level and from 0.76 to 0.80 at the domain level [23]. To assess quality of life in the present study, we used the Taiwanese version of the WHOQOL-BREF.

### 2.5. Statistical Analysis

We collected the following data from the medical records: age, sex, education, smoking status, drinking status, diagnosis, age at disease onset, BMI, BSRS-5, and quality of life. Data analysis was performed using Statistical Package for Social Sciences software (version 20.0; SPSS Inc., Chicago, IL, USA). The descriptive statistics examined were means, standard deviations (SDs), and percentages. Pearson’s correlation was performed to explore the relationships among demographic characteristics, BSRS-5 scores, and quality of life. A hierarchical linear regression model was used to evaluate significant risk factors for quality of life. The significance level was set at 0.05 using a two-tailed test for all analyses.

## 3. Results

The patients’ mean age was 24.72 years (SD = 3.51; Table 1). The vast majority (96%) were men, and most (65.3%) of them had more than a university education. The majority were non-smokers (79%) and non-drinkers (93.5%). The mean BMI was 23.92 (SD = 3.69). In total, 69.4% had a mood disorder diagnosis, with the mean age at disease onset being 22.48 years (SD = 4.96). The patients’ mean scores on the BSRS-5 and the Quality of Life were 24.72 (SD = 3.51) and 51.84 (SD = 12.56), respectively.

### 3.1. Correlations among Variables

The categorical variables about sex were significantly associated with quality of life (t = −2.61, *p* < 0.05; Table 2). As for the continuous variables, age (−0.20, *p* < 0.05), age at disease onset (−0.20, *p* < 0.05), and psychological symptoms (on the basis of the BSRS-5; −0.30, *p* < 0.01) were significantly associated with quality of life.

### 3.2. Hierachical Linear Regression for Quality of Life

Table 3 presents the inferential statistics of the outcomes. Across all hierarchical linear regression analyses, only sex was consistently significantly associated with quality of life in all models. In model 1, we included BMI, sex, age, education level, smoking status, and drinking status as independent variables of quality of life. The significant variable was sex (β = 0.21, 0.22, 0.19, *p* < 0.05). In model 2, we added approximate age at disease onset and diagnosis (R^2^ = 0.11 and ΔR² = 0.02). In model 3, the primary independent variable of psychological symptoms (according to the BSRS-5) was significant (β = −0.3, *p* < 0.01; R² = 0.18, ΔR² = 0.07).

### 3.3. Correlations between Items on the BSRS-5 and the Quality of Life with Total Score

As shown in Table 4, the BSRS-5 item ‘trouble falling asleep’ was significant (*p* < 0.05). Furthermore, the items ‘feeling tense or keyed up’, ‘feeling easily annoyed or irritated’, ‘feeling blue or sad’, and ‘feeling inferior to others’ were all significant (*p* < 0.01). The items in the physical and environmental domains of quality of life were significant (*p* < 0.01), as were the items in the psychological and social domains (*p* < 0.05).

## 4. Discussion

To our knowledge, this study is the first to explore the relationship between psychological symptoms and quality of life in patients with mood or adjustment disorders. Correlations were found between items on the BSRS-5 and quality of life with total score. Our study demonstrates that BSRS-5-measured psychological symptoms may be a key determinant of quality of life.

Psychological patients require long-term medical treatment, but poor compliance with medications leads to poor disease control, which affects quality of life. In this study, the psychological symptoms measured using the BSRS-5 meaningfully predicted scores in the physical, psychological, social, and environmental domains in patients with mood disorders. High scores on the BSRS-5 indicate poor quality of life.

Regarding correlations of sociodemographic and disease characteristics with psychological symptoms, gender and quality of life were correlated in our study. The previous studies on healthy workers demonstrated that the BSRS-5 scores of regular factory workers is predictive of scores in all four domains and on all 28 items of the Taiwanese version of the WHOQOL-BREF [7]. Huang, Chen [12] observed that in patients with CMI living in Kaohsiung City, disease factors were the most important predictors of quality of life. Similarly, psychological symptoms and age at disease onset were identified as predictors of quality of life in the present study. The additional sixth question provides the opportunity for the early detection of suicide risk. BSRS-5 for suicide prevention in Taiwan is recommended for administration to the general public [14,24].

Our findings suggest that psychological symptoms, as measured by BSRS-5, are important factors affecting quality of life. Randomized controlled trials can be used to make advances in nonpharmaceutical interventions for mitigating psychological symptoms in patients with mood disorders. The BSRS-5 scores of patients with mood or adjustment disorder are predictive of scores in all four domains and on all 28 items of the Taiwanese version of the WHOQOL-BREF.

## 5. Conclusions

Our findings revealed that that psychological symptoms, as measured by BSRS-5, are important determinants of quality of life in hospitalized patients suffering from mood and anxiety disorders. This study suggests that BSRS-5, a self-report questionnaire with a minimal number of items that patients can easily answer, is predictive of quality of life of hospitalized psychiatric patients.

## Figures and Tables

**Table 1 ijerph-19-00732-t001:** Demographic characteristics, disease characteristics, and psychological symptoms of the study population (*n* = 124).

Variables	*n* (%)	Mean (SD)
Demographics		
BMI		23.92 (3.69)
Age		24.72 (3.51)
Sex		
Female	5 (4.0)	
Male	119 (96.0)	
Education		
Bachelor’s degree or higher	81 (65.3)	
Associate degree or lower	43 (34.7)	
Smoking		
Yes	26 (21.0)	
No	98 (79.0)	
Drinking		
Yes	8 (6.5)	
No	116 (93.5)	
Disease characteristics		
Age at onset		22.48 (4.96)
Diagnosis		
Mood disorder	86 (69.4)	
Adjustment disorder	38 (30.6)	
Psychological symptoms		
BSRS-5		24.72 (3.51)
Quality of life		51.84 (12.56)

Note: BMI, body mass index; BSRS-5, Brief Symptom Rating Scale; SD, standard deviation.

**Table 2 ijerph-19-00732-t002:** Correlations among demographic characteristics, disease characteristics, psychological symptoms, and quality of life in patients with mood or adjustment disorder (*n* = 124).

Variable	Mean	SD	*t* Value	^†^ *p*
Sex			−2.61	0.01 *
Female	37.80	16.66		
Male	52.43	12.09		
Education			1.17	0.24
Bachelor’s degree or higher	50.88	12.70		
Associate degree or lower	53.65	12.23		
Smoking			−1.25	0.21
Yes	54.58	13.16		
No	51.11	12.36		
Drinking			−0.79	0.43
Yes	55.25	13.41		
No	51.60	12.52		
Diagnosis			0.31	0.76
Mood disorder	52.07	12.64		
Adjustment disorder	51.32	12.52		
Variable			**Correlation**	^ **‡** ^ * **p** *
Age			−0.20	0.03 *
BMI			−0.00	0.51
Age at onset			−0.20	0.05 *
BSRS-5			−0.30	0.00 **

Note: BMI, body mass index; BSRS, Brief Symptom Rating Scale; SD, standard deviation. ^†^
*p* = independent *t* test for categorical variables. ^‡^
*p* = Pearson correlation for continuous variables. * *p* < 0.05; ** *p* < 0.01.

**Table 3 ijerph-19-00732-t003:** Hierarchical linear regression model of quality of life (*n* = 124).

Variables	Model 1		Model 2		Model 3	
	Standardized β	*p* Value	Standardized β	*p* Value	Standardized β	*p* Value
Demographics						
BMI	−0.01	0.91	−0	0.99	−0.06	0.55
Age	−0.17	0.12	−0.10	0.49	−0.07	0.62
Sex						
Female	Ref.	-	Ref.	-	Ref.	-
Male	0.21	0.03	0.22	0.02	0.19	0.04
Education						
Bachelor’s degree or higher	0.02	0.89	0.04	0.71	0.10	0.36
Associate degree or lower	Ref.	-	Ref.	-	Ref.	-
Smoking						
Yes	0.04	0.67	0.06	0.57	0.07	0.47
No	Ref.	-	Ref.	-	Ref.	-
Drinking						
Yes	0.06	0.49	0.08	0.39	0.07	0.40
No	Ref.	-	Ref.	-	Ref.	-
Disease characteristics						
Age at onset	-	-	−0.15	0.23	−0.17	0.17
Diagnosis						
Mood disorder	-	-	Ref.	-	Ref.	-
Adjustment disorder	-	-	−0.07	0.47	0.04	0.68
Psychological symptoms						
BSRS	-	-	-	-	−0.30	0.00
R²	0.09		0.11		0.18	
ΔR²			0.02		0.07	

Note: BMI, body mass index; BSRS, Five-Item Brief Symptom Rating Scale.

**Table 4 ijerph-19-00732-t004:** Correlations between items on the BSRS-5 and Quality of Life with total score (*n* = 124).

		Total Quality of Life Score
BSRS-5 Items	Mean (SD)	Correlation
Trouble falling asleep (insomnia symptom)	1.74 (1.18)	−0.22 *
Feeling tense or keyed up (anxiety)	1.60 (1.11)	−0.24 **
Feeling easily annoyed or irritated (hostility)	1.48 (1.09)	−0.29 **
Feeling blue or sad (depression)	1.64 (1.15)	−0.28 **
Feeling inferior to others (interpersonal sensitivity)	1.37 (1.14)	−0.29 **
Suicide ideation ^$^	0.40(0.73)	−0.73
		Total BSRS-5 score
Quality of Life items	Mean (SD)	Correlation
Physical domain	6.85 (1.50)	−0.43 **
Psychological domain	7.16 (1.90)	−0.18 *
Social domain	7.73 (2.37)	−0.19 *
Environmental domain	7.89 (2.40)	−0.23 **

Note: BSRS-5, Five-Item Brief Symptom Rating Scale; SD, standard deviation. ^$^ Suicidal ideation was derived from the sixth extend item of BSRS-5. * *p* < 0.05; ** *p* < 0.01.

## Data Availability

The data presented in this study are available on request from the corresponding author. The data are not publicly available due to privacy.

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
