# Peer review of "Correlations between Psychiatric Symptoms and Quality of Life in Patients with Psychological Disorders: Hospital-Based Retrospective Study"

_ijerph, 2022, doi:10.3390/ijerph19020732_

Round 1

Reviewer 1 Report

My biggest concern about the manuscript is that the discussion and the conclusion are overestimated. 

The author found weak correlations between BSRS and quality of life, also the inclusion of BSRS at the prediction model only added 7% to the prediction in relation to the previous one. Moreover, the final model explained only 18% of the variability of the quality of life measured by the WHOQOL-BREF. It would be adequate to discuss the real applicability of the correlations and the prediction found, with a more realistic interpretation of the strength and the applicability of them. 

I am not sure that the 6th item added by the authors to the BSRS should be assumed as an item of the questionnaire itself. It could be handled as a separeted aspect assessed by the authors without compromising the validity of the original questionnaire. 

Author Response

Dear Editor and Reviewer’s
Manuscript ID: ijerph-1438250
Manuscript Title: “Correlations between Psychiatric Symptoms and Quality of Life in Patients with Psychological Disorders: Hospital-based retrospective study”
We sincerely thanks to the editor and all reviewers for their valuable suggestions and giving us great opportunity to revised our manuscript entitled “Correlations between Psychiatric Symptoms and Quality of Life in Patients with Mood Disorders: Hospital-based retrospective study”. We have incorporated all the suggested changes into the manuscript and have highlighted the revised sections. Hereby, our responses and revision based on editor and reviewer’s comments.
--------------------------------------------------------------

REVIEWER #1 COMMENTS:
1. It would be adequate to discuss the real applicability of the correlations and the prediction found, with a more realistic interpretation of the strength and the applicability of them.

Response: Thank you for your corrections and suggestions. Hereby we present the applicability of them in conclusions section.
“Clinically, patients with mood disorders presents with symptoms that affect their quality of life. Health care professionals can use BSRS-5 to measure psychological symptoms among these patients. This can assist in of early provision of intervention programs for those with problematic domains to help them return to the community and have a better quality of life.” (Line 244-248, Page 7).

2. I am not sure that the 6th item added by the authors to the BSRS should be assumed as an item of the questionnaire itself. It could be handled as a separeted aspect assessed by the authors without compromising the validity of the original questionnaire.

Response: Thank you for your corrections and suggestions. Hereby we present the ways to use in our study.
“We did separate the 6th item of BSRS questionaire to explore the relevance to QOL in Table 4. We presented 1th to 5th the total scores of BSRS questionaire in Tables 1 to 3. (Page 4-6).

Reviewer 2 Report

  1. Abbreviation of World Health Organization Quality of Life should appear in line 19 prior appearing as WHOQOL in line 25.
  2. The term ‘psychological disorder’ covers a broad spectrum of disorders that impact multiple areas of life. This study was specifically conducted on patients with mood disorders. As the abstract contains the summary of the entire study, authors may consider replacing the ‘psychological disorders’ word by ‘mood disorders’ in line 15, in order to avoid any misconceptions for the readers.
  3. Punctuation is missing in line 99.
  4. Could authors specify the age range?
  5. It is not clear if the patients included in this study were under medications? Could authors demonstrate if the medications can bias the analyses or findings of the study?
  6. Was there any specific reason to include less female participants as compared to the males?
  7. ‘Sex’ was found to be significantly correlated with mood or adjustment disorders. I wonder how reliable the finding is, as the number of females was very few (N = 5/124). Could authors provide a rationale on this?   

Author Response

Dear Editor and Reviewer’s
Manuscript ID: ijerph-1438250
Manuscript Title: “Correlations between Psychiatric Symptoms and Quality of Life in Patients with Psychological Disorders: Hospital-based retrospective study”

We sincerely thanks to the editor and all reviewers for their valuable suggestions and giving us great opportunity to revised our manuscript entitled “Correlations between Psychiatric Symptoms and Quality of Life in Patients with Mood Disorders: Hospital-based retrospective study”. We have incorporated all the suggested changes into the manuscript and have highlighted the revised sections. Hereby, our responses and revision based on editor and
reviewer’s comments.
------------------------------------------------------------------------------------

REVIEWER #2 COMMENTS:
1. Abbreviation of World Health Organization Quality of Life should appear in line 19 prior appearing as WHOQOL in line 25.

Response: Thank you for your valuable suggestion. We modified and highlighted in line 19 and 25. 

“the Taiwanese version of the abbreviated World Health Organization Quality of Life Questionnaire (WHOQOL-BREF).” (Line 19-20, Page 1).

;“This suggests that psychometrically measured psychological symptoms constitute critical determinants of WHOQOL.” (Line 26, Page 1).

2. The term ‘psychological disorder’ covers a broad spectrum of disorders that impact multiple areas of life. This study was specifically conducted on patients with mood disorders. As the abstract contains the summary of the entire study, authors may consider replacing the ‘psychological disorders’ word by ‘mood disorders’ in line 15, in order to avoid any misconceptions for the readers.

Response: Thank you for your valuable suggestion. We modified and highlighted in line 16, 35, 66. 

“The purpose of this study was to investigate potential effects of psychological symptoms on quality of life in patients with mood disorders.” (Line 16 35, 66). 

3. Punctuation is missing in line 99.

Response: Thank you for your valuable suggestion. We marked the punctuation in the line 99.
“(as indicated by scores on the World Health Organization Quality of Life (WHOQOLinstrument) were obtained from the medical record.” (Line 101-103, Page 3). 

4. Could authors specify the age range?

Response: Thank you for your valuable suggestion. We changed the word and noted the explanation in brackets.
 “We included patients older than 20 years with a diagnosis of mood or adjustment disorder using a convenience sampling method.” (Line 86-87, Page 2).

5. It is not clear if the patients included in this study were under medications? Could authors demonstrate if the medications can bias the analyses or findings of the study?

Response: Thank you for your valuable suggestion. We explained the limitations of the study in discussion section.
“This study has limitations that need to be considered. Firstly, patients included in this study may take medication such as mood stabilizer. Hofmann [25] conducted meta-analysis about the effect of different treatments of cognitive-behavioral therapy (CBT) and selective serotonin reuptake inhibitors (SSRIs) on the quality of life. The result showed significantly greater improvement in QOL using CBT, unlike SSRIs.” (Line 229-233, Page 7). 

6. Was there any specific reason to include less female participants as compared to the males?

Response: Thank you for your valuable suggestion. We explained the limitations of the study in discussion section.
“Another limitation was regarding sex. The hospital had more male patients than females, hence we listed sex as a variable to explore the correlation between BSRS and QOL by con-trolling it. Despite these limitations, our study found that each of BSRS-5 factor and each domain of QOL showed significant results.” (Line 233-237, Page 7).

7. ‘Sex’ was found to be significantly correlated with mood or adjustment disorders. I wonder how reliable the finding is, as the number of females was very few (N = 5/124). Could authors provide a rationale on this?

Response: Thank you for your valuable suggestion. We writed the reason why there were fewer female in the discussion section.
“Another limitation was regarding sex. The hospital had more male patients than females, hence we listed sex as a variable to explore the correlation between BSRS and QOL by con-trolling it. Despite these limitations, our study found that each of BSRS-5 factor and each domain of QOL showed significant results.” (Line 233-237, Page 7).

Reviewer 3 Report

1 Readers do not know the shortcomings of previous studies and your contributions.

2 What are the aim and research questions of this study?

3 You should give more evidence to motivate your study.

4 I suggest that you should reorganize the section 2 from the perspective of data, method and variables.

5 Conclusions section is to simple. You’d better to give more information on your whole study.

Author Response

Dear Editor and Reviewer’s
Manuscript ID: ijerph-1438250
Manuscript Title: “Correlations between Psychiatric Symptoms and Quality of Life in Patients with Psychological Disorders: Hospital-based retrospective study”
We sincerely thanks to the editor and all reviewers for their valuable suggestions and giving us great opportunity to revised our manuscript entitled “Correlations between Psychiatric Symptoms and Quality of Life in Patients with Mood Disorders: Hospital-based retrospective study”. We have incorporated all the suggested changes into the manuscript and have highlighted the revised sections. Hereby, our responses and revision based on editor and reviewer’s comments.
-----------------------------------

REVIEWER #3 COMMENTS:
1. Readers do not know the shortcomings of previous studies and your contributions.

Response: Thank you for your valuable suggestion. We rewrited the shortcomings of the previous research in the introduction section. 

“Studies have examined BSRS-5 related factors in elderly in-patients with surgical or medical conditions. However, they did not explore the mental factors related to the suicidal ideation [11]. Some studies included subjects from community residents with an exclusion of cases of mental illness [15]. Regarding research methods, some studies reported the use of telephone interviews as research methods. In these studies, psychiatric medical staff were not included as researchers to avoid bias [14, 15]. In addition, other studies indicated that apart from psychiatric patients, they also examined adolescents, suspected or confirmed coronavirus disease-19 (COVID-19) patients, healthcare workers
and soldiers [14, 16, 17, 18].” (Line 70-178, Page 2).

2. What are the aim and research questions of this study?

Response: Thank you for your valuable suggestion. We explained the aim and the research questions in the introduction section.

“Overall, previous studies have rarely explored the correlation between BSRS and quality of life among patients with mood disorder or adjustment disorder. In this study, the aim was to investigate the potential impact of psychological symptoms on quality of life of patients with mood disorder or adjustment disorder using self-report questionnaires. Predicting risk factors helps in early detection and provision of appropriate intervention programs.” (Line 79-84, Page 2).

3. You should give more evidence to motivate your study.

Response: Thank you for your valuable suggestion. We re-examined the previous
research and present the shortcomings one by one to increase the motivation of our research in the introduction section.

“Studies have examined BSRS-5 related factors in elderly in-patients with surgical or medical conditions. However, they did not explore the mental factors related to the suicidal ideation [11]. Some studies included subjects from community residents with an exclusion of cases of mental illness [15]. Regarding research methods, some studies reported the use of telephone interviews as research methods. In these studies, psychiatric medical staff were not included as researchers to avoid bias [14, 15]. In addition, other studies indicated that apart from psychiatric patients, they also examined adolescents, suspected or confirmed coronavirus disease-19 (COVID-19) patients, healthcare workers
and soldiers [14, 16-18].” (Line 70-178, Page 2). 

4. I suggest that you should reorganize the section 2 from the perspective of data, method and variables.

Response: Thank you for your valuable suggestion. We used the title to distinguish.

”2.1 Participants and settings (Line 86, Page 2).
2.2 Five-Item Brief Symptom Rating Scale (BSRS-5) (Line 104, Page 3).
2.3 The abbreviated World Health Organization Quality of Life Questionnaire
(WHOQOL-BREF) (Line 122, Page 3).
2.4 Statistical analysis” (Line 144, Page 3).

5. Conclusions section is to simple. You’d better to give more information on your whole study.

Response: Thank you for your valuable suggestion. We explained from the clinical application in the conclusion section.

“This study confirms that BSRS-5 scores predict the scores in all 4 domains and on all 28 items of the Taiwanese version of the WHOQOL-BREF in patients with mood disorders. BSRS-5 is a self-report questionnaire that has a small number of questions and is easy to answer. In addition, BSRS-5 has shown to be an efficient screening tool to assess patients with mood disorders” (Line 239-243, Page 7).
“Clinically, patients with mood disorders presents with symptoms that affect their quality of life. Health care professionals can use BSRS-5 to measure psychological symptoms among these patients. This can assist in of early provision of intervention programs for those with problematic domains to help them return to the community and have a better quality of life.” (Line 244-248, Page 7)